# Evaluation on the Growth Performance, Nutrient Digestibility, Faecal Microbiota, Noxious Gas Emission, and Faecal Score on Weaning Pigs Supplement with and without Probiotics Complex Supplementation in Different Level of Zinc Oxide

**DOI:** 10.3390/ani13030381

**Published:** 2023-01-22

**Authors:** Huan Wang, Shi-Jun Yu, In-Ho Kim

**Affiliations:** 1School of Biology and Food Engineering, Chuzhou University, Chuzhou 239012, China; 2Department of Animal Resource & Science, Dankook University, Cheonan 31116, Republic of Korea

**Keywords:** probiotics, zinc oxide, growth performance, weaning pigs

## Abstract

**Simple Summary:**

Long-term use of pharmacological levels of ZnO has caused environmental pollution and bacterial resistance. We mainly study the effect of adding probiotics complex supplementation with high and low ZnO diets on the performance of weaning pigs in 42 days. The results: probiotic supplementation has reduced the fecal *Escherichia coli* counts. There were no interactive effects between ZnO and probiotic complex supplementation on all the measured parameters. In the current study, supplementation of 0.1% probiotic complex to 300 ppm ZnO diet did not show the same benefit with pharmacological doses of ZnO.

**Abstract:**

A total of 200 26-day-old crossbred weaning piglets ((Yorkshire × Landrace) × Duroc; 6.55 ± 0.62 kg) were used in a 6-week experiment to evaluate the effects of adding probiotics complex supplementation (Syner-ZymeF10) with high and low ZnO diets on the performance of weaning pigs in 42 days. Pigs were randomly allotted to a 2 × 2 factorial arrangement and they were supplemented with two concentration level of ZnO with 3000 ppm and 300 ppm and probiotics complex supplementation with 0 and 0.1%. There were ten replicate pens per treatment with five pigs per pen (two gilts and three barrows). Pigs fed diets with 3000 ppm ZnO had a higher BW during the overall period and ADG during d 8–21, d 22–42, and overall period than pigs receiving 300 ppm ZnO diets (*p* < 0.05), as well as a G: F which tended to increase on d 8–21 and overall period (*p* < 0.1) and decreased tendency on faecal gas emission of methyl mercaptans and acetic acid concentration (*p* < 0.1). Dietary probiotics complex supplementation had decreased the *E. coli* count (*p* < 0.05) and tended to increase the *Lactobacillus* count (*p* < 0.1). Dietary probiotics complex supplementation and different level of ZnO supplementation had no significant effect on the nutrition digestibility and faecal score (*p* > 0.05). In conclusion, probiotic supplementation reduced the fecal *E. coli* counts and tended to improve *Lactobacillus* counts. There were no interactive effects between ZnO and probiotic complex supplementation on all the measured parameters.

## 1. Introduction

Birth and weaning of piglets will have major modifications on intestine changes [1]. Weaning is a very challenging and stressful period in the life of piglets due to the switch in the form of feed from liquid to solid, changing to a new living environment and touching other new piglets, which results in the dramatic changes to the gut microbiota, especially within 1 to 2 weeks after weaning [2,3]. These stresses will cause diarrhea and growth retardation in piglets, which affect the production efficiency of the pigs and result in serious economic losses [4]. Previously, Milani et al. [5] emphasized the ways to overcome weaning stresses with the inclusion of antibiotics and mineral compounds, such as zinc oxide (ZnO) supplements in the diet.

However, the use of antibiotics in feed has been banned in the European Union since 2006 due to side effects such as the evolution of resistant strains of bacteria and their residues in meat and meat products [6]. In South Korea, antibiotics used for growth promoting in animal feed were also banned in July 2011 [7]. Zinc is generally considered as essential element for life and plays an important role as a catalytic and structural cofactor in cell metabolism as a divalent cation (Zn^2+^) [8]. Deficiency of zinc can impair overall immune function and resistance to infection [9]. Pharmacological levels of ZnO (3000 ppm) have been used by the livestock industry in weaned piglet diets to promote growth and deal with diarrhea problems. However, the absorption rate of ZnO in animals is low, and about 80% is excreted through faeces [10]. Long-term use of pharmacological levels of ZnO has caused environmental pollution and bacterial resistance [11,12]. In view of the potential negative impact of ZnO, the European regulations limited the use of Zn content in pig diets up to 150 mg/kg, which is far below than pharmacological level [13]. However, most national standard practices in Asia and the Americas still use 3000 ppm ZnO in weaning pigs [14].

Probiotics are non-pathogenic microorganisms that can provide health benefits to the host when ingested in sufficient amounts [15], the positive effects of probiotics have been demonstrated to promote intestinal microflora balance and improve growth performance in weaning pigs [16,17,18] and growing and finishing pigs [19,20,21].

The probiotics complex is reported to have better therapeutic effects than a single species, and single species often exposed to limited functionality [22]. In many previous research reports, the probiotics complex has also shown good probiotic effects. Liu et al. [23] demonstrated that dietary complex probiotic supplementation (*B. subtilis* and *S. cerevisiae*) in diet improved growth performance and some amino acids. Similarly, research by Lan et al. [24] showed that supplementation of complex probiotics in low nutrient-dense diets has shown improved related growth performance in weaning pigs. In this research, we hypothesized that dietary supplementation with probiotics complex has a potential substitution effect on ZnO. Thus, the present study was conducted to evaluate the effect of probiotics complex supplementation with high and low level of ZnO diet with an objective to figure out the possibility of reducing ZnO supplementation by adding probiotic complex to the diet on the growth performance, nutrient digestibility, faecal microbiota, noxious gas emission and faecal score of weaning pigs.

## 2. Material and Methods

The experimental protocol used in this study was reviewed and approved by the Animal Care and Use Committee (Dankook University DK-1-1937).

### 2.1. Source of Probiotics

Our research used the probiotics complex (SynerZymeF10), which was kindly provided by SynerBig Co. Ltd. (Seoul, Korea). The probiotics complex was a mixture of spray-dried spores and contained the *Bacillus subtilis* (1 × 10^12^ CFU·kg^−1^), *B. coagulans* (1 × 10^12^ CFU·kg^−1^), *B. licheniformis* (5 × 10^11^ CFU·kg^−1^) and *Clostridium butyricum* (1 × 10^11^ CFU·kg^−1^).

### 2.2. Experimental Design, Animals and Housing

A total of 200 26-day-old crossbred weaning piglets ((Yorkshire × Landrace) × Duroc; 6.55 ± 0.62 kg) were used in the experiment (6 weeks). Weaning piglets were randomly allotted to a 2 × 2 factorial arrangement and they were supplemented with two concentration levels of ZnO with 3000 ppm and 300 ppm and probiotics complex supplementation with 0 and 0.1%. There were 10 replicate pens per treatment with 2 gilts and 3 barrows per pen, and all pigs were divided fed into three phases: d 1–7 (phase 1), d 8–21 (phase 2), and d 22–42 (phase 3). All diets were formulated to meet or exceed the National Research Council (NCR) [25] nutrient requirements (Table 1). Pigs were housed in an environmentally controlled nursery room with a slatted plastic floor. During the whole experiment period, all weaning piglets were provided *ad libitum* access to feed and water through a self-feeder and nipple drinker, respectively. The environmental temperature and humidity were kept at 30 °C from d 1 to 7 and 60%, respectively, and lowered by 1 °C per week.

### 2.3. Growth Performance and Nutrient Digestibility

Beginning on days 1, 7, 21, and 42 of the experiment, record individual pig body weight (BW) and feed consumption on the basis of the pen. Average daily gain (ADG), average daily feed intake (ADFI), and gain ratio (G:F) were calculated accordingly.

Pigs were offered diets with chromic oxide (3 g·kg^−1^) as an indigestible marker to determine the apparent total tract digestibility (ATTD) of dry matter (DM), nitrogen (N), and gross energy (GE). On d 35, the faecal samples of per weaning pig were collected via rectal massage and the same pen samples were pooled and mixed, then the feed and faeces samples were stored at −20 °C immediately and kept until analysis in the laboratory. All samples were analyzed for DM (method 930.15) and N (method 984.13) according to the standard procedures of the AOAC [26]. Chromium was analyzed via UV absorption spectrophotometry (UV-1201, Shimadzu Corp., Kyoto, Japan) according to the method described by Williams et al. [27]. GE was determined by measuring the heat of combustion in the samples, using a Parr 6100 oxygen bomb calorimeter (Parr instrument Co., Moline, IL, USA).

The ATTD was then calculated using the following formula:$$ATTD\;\left(\%\right)=\left.\left\{1-\left[\frac{\left(Nf\times Cd\right)}{\left(Nd\times Cf\right)}\right]\right.\right\}$$
where Nf = nutrient concentration in feces (% DM), Cd = chromium concentration in diet (% DM), Nd = nutrient concentration in diet (% DM), and Cf = chromium concentration in feces (% DM).

On d 42, fresh faecal samples were collected directly via massaging the rectum of the weaning pigs in each pen (1 gilt and 1 barrow), and then samples (kept on the ice pack) were passed to the laboratory for microbiota and noxious gas emissions analysis immediately, according to the method described by Sun and Kim [28]. One gram of faecal microbial flora from each pen was diluted with 9 mL of 1% peptone broth (Becton, Dickinson) and homogenized. Then, viable counts of bacteria were determined by plating serial 10-fold dilution (in 1% peptone solution) onto MacConkey agar plates (Difco Laboratories, Detroit, MI, USA; 37 °C, 24 h) and lactobacilli medium III agar plates (Medium 638; DSMZ, Braunschweig, Germany; 39 °C, 48 h) to isolate the *E. coli* and *Lactobacillus*. All bacterial counts were reported as log10 colony-forming units per gram. On d 7, 21, and 42, we observed and recorded the fecal score of each pig based on fecal consistency: 1, hard, dry pellets in a small, well-firmed feces; 2, slightly soft, formed stool that remains firm and soft; 3, soft, formed, and moist stool that retains its shape; 4, loose and semi-liquid, unformed stool that assumes the shape of the container, and 5, watery, liquid stool that can be poured.

Fresh fecal samples (300 g) were stored in 2.6 L plastic boxes and fermented in an incubator at the set time and temperature (30 h, 35 °C). Then, the NH_3_, H_2_S, methyl mercaptan, acetic acid, and CO_2_ were measured using a complex gas meter (MultiRAE Lite model PGM-6208, RAE, USA).

### 2.4. Statistical Analyses

All data were analyzed by analysis of variance using the GLM procedure of SAS as a randomized 2 × 2 factorial design (SAS Institute Inc., Cary, NC, USA) and the pen was used as an experimental unit. The data were tested for the main effects of probiotics and ZnO, as well as the interaction between probiotics and ZnO. A *p* < 0.05 was considered to be statistically significant, whereas 0.10 > *p* > 0.05 was considered a trend.

## 3. Results

### 3.1. Growth Performance and Nutrient Digestibility

As presented in Table 2, weaning pigs fed high ZnO diets had a higher BW during the overall period (*p* < 0.05), and a higher ADG during d 8–21, d 22–42, and overall period than those with low ZnO diets (*p* < 0.05), as well as an increased tendency of G:F during d 8–21 and overall period (*p* < 0.1), whereas no difference in ADFI among all treatments throughout the experiment (*p* > 0.05). No probiotics effects or interactive effects of probiotic with ZnO was observed among treatments on growth performance (*p* > 0.05). The probiotics supplementation and different level of ZnO had no difference or interactive effects in the nutrient digestibility (*p* > 0.05).

### 3.2. Fecal Microbiota and Fecal Gas Emissions

As presented in Table 3, weaning pigs diet supplemented with probiotics complex decreased the *E. coli* count (*p* < 0.05) and meant an increased tendency of *Lactobacillus* (*p* < 0.1). The 3000 ppm ZnO diet exerted decreased tendency in the emission of methyl mercaptans and acetic acid concentration from faeces (*p* < 0.1). No probiotic and ZnO interactive effect were observed in the *E. coli* and *Lactobacillus* count or noxious gas emission (*p* < 0.05).

### 3.3. Fecal Score

As presented in Table 4, dietary probiotics complex supplementation and different level of ZnO supplementation made no difference or had an interactive effect on the faecal score.

## 4. Discussion

Piglets usually show reduced intake of feed and water after weaning which will cause atrophy of the intestinal villi and a decrease in digestion and absorption capacity. Weaning stress can also cause perturbations of the intestinal epithelium, weaken the immune system, and modify the intestinal flora, and, ultimately, affect the health and growth performance of the piglets and even mortality [29]. According to relevant research reports, the addition of a probiotics diet to weaning pigs with a pharmacological level of ZnO is regarded as an effective way to overcome or alleviate the effects of stress [30,31,32]. Probiotics supplementation in weaner pig diet has been proven by many studies to reduce weaning stress and diarrhea, as well as improve the growth performance and intestinal health [33,34]. However, the response of piglets to probiotic supplementation is variable due to the addition dosage, living environment, health status, and strain differences, etc. [35,36,37]. In our current study, probiotics complex supplementation did not show the significant effects on growth performance of weaning pigs. This finding is correlated with Nguyen et al. [38], who suggested that the dietary supplement 0.1% probiotics mixture with SynerZymeF10 did not significantly improve the growth performance of weaned piglets. Moreover, Min et al. [39] also reported that the addition of a dietary mixture of probiotics (Syncra^®^ SWI 201) in growing–finishing pig diets had no beneficial effects on growth performance. On the other hand, ZnO which contained 3000 ppm in the diet significantly increased ADG and overall BW compared to pigs fed 300 ppm of ZnO diet. This is consistent with previous reports on the application of ZnO in pharmacological dosage. According to the reports of Carlson et al. [40] and Upadhaya et al. [41], ZnO has been gradually used more in weaning piglets’ diet since 1990 with a pharmacological dosage and it has been used as an effective alternative to antibiotics. In the present study it was speculated that supplementing probiotics complex to low ZnO may contribute to the reduction of the negative impact of ZnO on the environment. However, we did not find the interactive effect on growth performance between probiotics and ZnO in pigs’ diets.

In a study by Wang et al. [42], 0.1% probiotics complex (*B. subtilis* and *B. licheniformis*) supplementation in diets did not lead to significant effects on DM and N digestibility in pigs. A similar result was observed by other researchers [43,44]. Interestingly, with a contrary result, Lan et al. [45] noted that the addition of a probiotics complex (SynerZymeF10) diet to weaned pigs showed that 0.1% level probiotics complex can increase the ATTD of DM, N, and GE, but the levels of 0.01%, 0.03%, and 0.06% did not exert significant effects. We analyzed the possible causes for the variations on the results of this study, such as diet composition, feed form, probiotic added levels, and pig age. In this study, probiotics were added to the ZnO diet but lacked a basic diet as a control, so was also limited. In the current study, there was no difference on nutrient digestibility of DM, N, and GE. This could explain the lack of effect resulting from the probiotics complex on growth performance of weaning pigs. Meanwhile, our result indicated that the 3000 ppm ZnO exerted beneficial growth performance compared with 300 ppm ZnO, which may have been due to the increase in G: F, but not in the ATTD. It is reported that zinc can increase the secretion of ghrelin by the stomach, thereby stimulating the secretion of growth hormone and insulin [46].

The intestine microbiota consists of hundreds of bacterial species and plays an important role in the health of the host [47]. It is reported that the probiotics can stabilize and restore the microbial equilibrium in the gut and can directly affect microorganisms, namely commensal and pathogenic ones [48]. In the weaning period, probiotics can positively influence the intestinal epithelium integrity, appropriate maturation of the gut associated tissue, and function of the neuro-endocrine system [49]. Nowadays, how probiotics regulate intestinal flora is still a research hotspot. In the current study, supplementing the diets with probiotics complex had a significant effect on *E. coli* count and the *Lactobacillus* count tended to increase. This result suggests that probiotics complex (*B. coagulans, B. licheniformis, B. subtilis,* and *C. butyricum*) supplementation was beneficial for regulation of intestinal flora, which inhibits the increase of *E. coli* in the gut of the host animal. In agreement with our study, Hu et al. [50] suggested that 2 × 10^9^ or 4 × 10^9^ CFU/kg *B. subtilis* KN-42 supplementation decreased the fecal *E. coli* counts and had no significant effects on the fecal *Lactobacillus* counts in weaned piglets. In pigs, the results of studies are inconsistent. Dong et al. [16] indicated that the probiotics complex containing *B. subtilis B27* and *Lactobacillus plantarum* GF103 (1.0 × 10^8^ CFU/kg and10^3^ 4.3 × 10^9^ CFU/kg, respectively,) had no effect on fecal *Lactobacillus* and *E. coli* counts in weaned piglets. Menegat et al. [38] also reported no significant differences in faecal *Lactobacillus* count of nursing piglets fed a *Bacillus subtilis* C-3102-supplemented diet compared with the basal diet; this may also be related to the lack of impact on growth performance. Moreover, Balasubramanian et al. [51] demonstrated that the diets supplemented with probiotics complex (*B. coagulans*, *B. licheniformis*, *B. subtilis* and *C. butyricum*) had increased faecal *Lactobacilli* and decreased *E. coli* counts in growing–finishing pigs during the entire experiment. The inconsistent results from previous studies may be associated with pig growth phase, strains of probiotic, and the composition of probiotics complex. In addition, we did not find an effect on faecal *Lactobacilli* and *E. coli* counts among the ZnO diets, and, additionally, there were no interactive effects observed on fecal microbial counts with probiotics and ZnO diet. The supplementation of probiotics or high levels of ZnO (3000 ppm) reduced the diarrhea of weaning pigs, as was proved by Ou et al. [52] and Pan et al. [53]. In our study, diarrhea in piglets was not observed in any treatment groups, because of the good feeding environment.

Pig are one of the biggest groups of livestock in the world, and the odorous gas emissions, such as NH_3_, H_2_S, and total mercaptans, are major aerial pollutants originating from animal production [54,55]. The emission of harmful gases seriously threaten the health of humans and animals. Among these noxious gas emissions, ammonia is the major aerial pollutant. Supplementing probiotics reduces fecal ammonia emissions and is mainly related to nutrients utilization and intestinal microflora ecosystem [56]. In this experiment, although there was no statistical difference in ammonia emissions between each treatment, the treatments were all at a very low level, which can be explained by the following reasons. On one hand, the effects of probiotics on nutrient digestibility were not observed, but we found the concentration of *E. coli* was reduced and the tendency of *Lactobacillus* was to increase. On the other hand, zinc has diverse regulatory, metabolic, and structural functions [57]. The present study showed that high dose of ZnO exhibited a beneficial effect on ADG and G: F, which may be another reason for the low level of ammonia emissions.

## 5. Conclusions

In conclusion, the supplementation of probiotics complex (*B. coagulans*, *B. licheniformis*, *B. subtilis* and *C. butyricum*) in the diet significantly reduced the *E. coli* count and tended to increase the *Lactobacillus* count. The pharmacological doses of ZnO (3000 ppm) improved the growth performance compared to 300 ppm, but no interactive effect was observed. In the current study, the probiotic complex showed the potential to partially replace zinc oxide, and the detailed mechanism still needs follow-up experimental research.

## Figures and Tables

**Table 1 animals-13-00381-t001:** Composition of weaning pig diets. (as fed-basis).

Item	Phase1		Phase2		Phase3	
	High ZnO	Low ZnO	High ZnO	Low ZnO	High ZnO	Low ZnO
Ingredients (%)						
Corn	39.32	40.04	51.67	52.39	58.48	59.18
Soybean meal	16.22	16.10	16.74	16.62	22.60	22.48
Fermented soybean meal	5.00	5.00	4.00	4.00	3.00	3.00
Spray dried plasma protein	6.00	6.00	3.00	3.00	-	-
Tallow	2.82	2.56	2.82	2.56	2.77	2.53
Lactose	12.88	12.88	7.78	7.78	3.18	3.18
Sugar	3.00	3.00	3.00	3.00	3.00	3.00
Whey protein	11.00	11.00	7.00	7.00	3.00	3.00
Monocalcium phosphate	0.88	0.88	1.08	1.08	1.15	1.15
Limestone	1.18	1.18	1.20	1.20	1.22	1.22
Salt	0.20	0.20	0.10	0.10	0.10	0.10
Methionine (99%)	0.20	0.20	0.15	0.15	0.08	0.08
Lysine	0.49	0.49	0.65	0.65	0.61	0.61
Mineral mix ^1^	0.20	0.20	0.20	0.20	0.20	0.20
Vitamin mix ^2^	0.20	0.20	0.20	0.20	0.20	0.20
Choline (25%)	0.03	0.03	0.03	0.03	0.03	0.03
Zinc oxide (80%)	0.38	0.04	0.38	0.04	0.38	0.04
Calculated value						
Crude protein, %	20.00	20.00	18.00	18.00	18.00	18.00
Metabolizable energy, kcal/kg	3450	3450	3400	3400	3350	3350
Calcium, %	0.80	0.80	0.80	0.80	0.80	0.80
Phosphorus, %	0.60	0.60	0.60	0.60	0.60	0.60
Lysine, %	1.60	1.60	1.50	1.50	1.40	1.40
Methionine, %	0.48	0.48	0.40	0.40	0.35	0.35
Fat, %	4.52	4.28	4.91	4.67	5.14	4.93
Zinc oxide, ppm	3053	333	3054	334	3057	337

^1^ Provided per kg of complete diet: Fe, 100 mg as ferrous sulfate; Cu, 17 mg as copper sulfate; Mn, 17 mg as manganese oxide; I, 0.5 mg as potassium iodide; and Se, 0.3 mg as sodium selenite. ^2^ Provided per kg of complete diet: vitamin A, 10,800 IU; vitamin D3, 4000 IU; vitamin E, 40 IU; vitamin K3, 4 mg; vitamin B1, 6 mg; vitamin B2, 12 mg; vitamin B6, 6 mg; vitamin B12, 0.05 mg; biotin, 0.2 mg; folic acid, 2 mg; niacin, 50 mg; D-calcium pantothenate, 25 mg.

**Table 2 animals-13-00381-t002:** Effects of ‘SynerZymeF10’ supplementation in ZnO diets on growth performance and nutrient digestibility in weaning pigs ^1^.

Items	−Pro		+Pro		SEM ^2^	*p*-Value ^3^		
	High ZnO	Low ZnO	High ZnO	Low ZnO		Pro	ZnO	Pro × ZnO
BW, kg								
initial	6.55	6.55	6.55	6.55	0.20	0.998	0.998	0.998
finish	26.39	25.71	26.74	25.93	0.34	0.412	0.035	0.853
D 1–7								
ADG, g	166	166	162	169	3.95	0.860	0.355	0.651
ADFI, g	200	203	202	205	5.94	0.946	0.559	0.516
G: F	0.911	0.893	0.891	0.903	0.01	0.535	0.772	0.540
D 8–21								
ADG, g	410	400	417	396	7.56	0.859	0.046	0.483
ADFI, g	473	484	490	479	8.60	0.708	0.250	0.592
G: F	0.838	0.841	0.842	0.831	0.01	0.316	0.064	0.716
D 22–42								
ADG, g	604	592	624	612	10.07	0.241	0.025	1.000
ADFI, g	840	843	849	854	8.84	0.654	0.230	0.911
G: F	0.719	0.703	0.735	0.717	0.01	0.219	0.184	0.940
Overall								
ADG, g	461	459	474	464	7.21	0.359	0.019	0.826
ADFI, g	611	616	619	621	6.01	0.650	0.223	0.650
G: F	0.756	0.745	0.763	0.748	0.01	0.189	0.072	0.911
Nutrient digestibility, %								
Dry matter	81.47	80.67	82.40	81.14	0.81	0.778	0.214	0.398
Nitrogen	78.87	78.56	79.74	78.74	0.66	0.501	0.403	0.654
Gross energy	79.69	79.27	80.66	79.43	0.65	0.396	0.220	0.543

^1^ Abbreviation: Pro, probioic; −Pro, without probiotic (SynerZymeF10) supplementation; +Pro, with 0.1% probiotic (SynerZymeF10) supplemtntation; High ZnO diet, basal diet + 3000 ppm ZnO; Low ZnO diet, Basal diet +300 ppm ZnO. ^2^ Standard error of means. ^3^ Means in the same row with different superscripts differ (*p* < 0.05).

**Table 3 animals-13-00381-t003:** Effects of ‘SynerZymeF10’ supplementation in ZnO diets on fecal microbiota and gas emission in weaning pigs ^1^.

Items	−Pro		+Pro		SEM ^2^	*p*-Value ^3^		
	High ZnO	Low ZnO	High ZnO	Low ZnO		Pro	ZnO	Pro × ZnO
*E. coli* (log_10_CFU/g)	6.26	6.20	6.13	6.16	0.04	0.054	0.765	0.326
*Lactobacillus* (log_10_CFU/g)	9.15	9.17	9.24	9.23	0.04	0.099	0.917	0.808
Gas emission, ppm								
NH_3_	1.4	1.9	1.0	1.6	0.7	0.650	0.472	0.933
H_2_S	1.9	2.3	1.6	3.0	0.7	0.780	0.221	0.446
Methyl mercaptans	2.3	4.9	1.6	3.5	1.1	0.352	0.061	0.758
CO_2_	700	875	450	725	258	0.424	0.370	0.840
Acetic acid	0.5	0.9	0.2	0.7	0.2	0.259	0.072	0.956

^1^ Abbreviation: Pro, probioic; −Pro, without probiotic (SynerZymeF10) supplementation; +Pro, with 0.1% probiotic (SynerZymeF10) supplemtntation; High ZnO diet, basal diet + 3000 ppm ZnO; Low ZnO diet, Basal diet +300 ppm ZnO. ^2^ Standard error of means. ^3^ Means in the same row with different superscripts differ (*p* < 0.05).

**Table 4 animals-13-00381-t004:** Effects of ‘SynerZymeF10’ supplementation in ZnO diets on faecal score in weaning pigs ^1^.

Items	−Pro		+Pro		SEM ^2^	*p*-Value ^4^		
	High ZnO	Low ZnO	High ZnO	Low ZnO		Pro	ZnO	Pro × ZnO
D 7 ^3^	3.61	3.66	3.59	3.61	0.06	0.543	0.516	0.760
D 21 ^3^	3.43	3.45	3.39	3.46	0.11	0.922	0.704	0.819
D 42 ^3^	3.21	3.25	3.21	3.27	0.07	0.908	0.543	0.908

^1^ Abbreviation: Pro, probioic; −Pro, without probiotic (SynerZymeF10) supplementation; +Pro, with 0.1% probiotic (SynerZymeF10) supplemtntation; High ZnO diet, basal diet + 3000 ppm ZnO; Low ZnO diet, Basal diet +300 ppm ZnO. ^2^ Standard error of means. ^3^ fecal score: 1–5, where 1, hard, dry pellets in a small, well-firmed feces; 2, slightly soft, formed stool that remains firm and soft; 3, soft, formed, and moist stool that retains its shape; 4, loose and semi-liquid, unformed stool that assumes the shape of the container; 5, watery, liquid stool that can be poured. ^4^ Means in the same row with different superscripts differ (*p* < 0.05).

## Data Availability

The data presented in this study are available on request from the corresponding author. The data are not publicly available due to privacy or ethical restrictions.

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
