# Peer review of "Evaluation on the Growth Performance, Nutrient Digestibility, Faecal Microbiota, Noxious Gas Emission, and Faecal Score on Weaning Pigs Supplement with and without Probiotics Complex Supplementation in Different Level of Zinc Oxide"

_animals, 2023, doi:10.3390/ani13030381_

Round 1

Reviewer 1 Report

“Evaluation on the growth performance, nutrient digestibility, faecal microbiota, noxious gas emission, and faecal score on weaning pigs supplement with and without probiotics complex supplementation in different level of zinc oxide”

This MS needs to improve a lot for format of for this journal etc. loss of Simple Summary, Reference style and English writing.

Simple Summary: Disappear

Abstract: The author must review the contents of this section and clarify the purpose of this MS.

Introduction:

Line 65-71: Author have to explain more or cite more reference for evident to prove about Probiotic complex. In this MS, Author do not work on ratio or portion of Probiotic complex. And more clarified on the objective or hypothesis for this MS.

Material and method & Result:

1) Line 111-115: Method for “coefficient of total tract apparent digestibility (CTTAD)”, Please check and correct on the equation and unit.

2) In table 2 : unit of nutrient digestibility? I think it is a percentage, but in the equation of CTTAD, that was not a percentage.

3)Line 116-124: The author's method must be referenced.

4) Table 3 : need unit in E.coli and  Lactobacillus parameter.

Conclusion: Author have to improve to conclude the result that related to answer the objective.

Author Response

Response to Reviewer 1 Comments

Dear honoured reviewers,

We really appreciate you and the reviewers for your comments on our Manuscript: Evaluation on the growth performance, nutrient digestibility, faecal microbiota, noxious gas emission, and faecal score on weaning pigs supplement with and without probiotics complex supplementation in different level of zinc oxide We express our sincere gratitude and thankfulness for your time and precision in reviewing our manuscript. The responses to the comments are as follows. For your kind information, we have carefully dealt with the comments of the reviewers as follows: All the places which we changed have already been marked yellow in our paper, for the purpose of highlight. We hope the revised manuscript meets the standard of publication. Thank you!

Simple Summary:

Point 1: Simple Summary: Disappear

Response 1: Thanks for your suggestion, we have edited, please see line 12-18.

Abstract:

Point 2: The author must review the contents of this section and clarify the purpose of this MS.

Response 2: Thank you for your advice. We have checked purpose of this manuscript and modified in article, please see line 21-23.

Introduction:

Point 3: Line 65-71: Author have to explain more or cite more reference for evident to prove about Probiotic complex. In this MS, Author do not work on ratio or portion of Probiotic complex. And more clarified on the objective or hypothesis for this MS.

Response 3: Thank you very much for the comments and suggestions. We followed your suggestion, and then added in Lines 87-95.

Material and method & Result:

Point 4: Line 111-115: Method for “coefficient of total tract apparent digestibility (CTTAD)”, Please check and correct on the equation and unit.

Response 4: Thank you for your advice. We have checked and modified in article, please see line 140-145.

Point 5: In table 2 : unit of nutrient digestibility? I think it is a percentage, but in the equation of CTTAD, that was not a percentage.

Response 5: Thank you very much for the comments. We have checked and modified in article, please see Table 2.

Point 6: Line 116-124: The author's method must be referenced.

Response 6: Thanks for your suggestion, we have edited, please see line 149.

Point 7: Table 3 : need unit in E.coli and Lactobacillus parameter.

Response 7: Thanks for your suggestion, we have edited, please see Table 3.

Conclusion:

Point 8: Author have to improve to conclude the result that related to answer the objective.

Response 8: Thanks for your suggestion, we have edited, please see line 87-94, 304-306.

Reviewer 2 Report

This is a straight forward article and clearly presented. 

The following comments should be answered.

1. The experimental diet in phase 1 contained 20 % lactose while 12.88 % lactose was incorporated in the diet. What other sources of lactose were in the diet. I can imagine that it is the whey protein. If so, please mention the quality of both the lactose and the wheyprotein.

2. The authors do not motivate the choose for the days the fecal samples were taken. The majority of the diarrhea problems occur between 5 and 12 days postweaning. Yet, the digestibility samples and the samples for microbial analyses were obtained well after the critical period. They should motivate why they choose these periods. At day 35 and 42 p.w. the intestinal conditions are already stable. So one can expect that differences  will be less than in the critical period.

3. The diets fed are "luxureous" and highly digestible. In order to measure an interaction a more risky diet should have been used. Even an effect of high levels of ZnO on diarrhea was not observed. 

4. In my opinion in the discussion the authors could have paid more attention between the relation ADG and G:F D22-42 and the digestibility data. The feed efficiency and the digestibility data seem to be correlated. The bacteriostatic effect of ZnO had an effect on both. The addition of ZnO causes that less andogeneous protein is formed what probably caused the hgiher protein digestibility and G:F. 

5. I would not mention line 191-193

.

Author Response

Response to Reviewer 2 Comments

Dear honoured reviewers,

We really appreciate you and the reviewers for your comments on our Manuscript: Evaluation on the growth performance, nutrient digestibility, faecal microbiota, noxious gas emission, and faecal score on weaning pigs supplement with and without probiotics complex supplementation in different level of zinc oxide We express our sincere gratitude and thankfulness for your time and precision in reviewing our manuscript. The responses to the comments are as follows. For your kind information, we have carefully dealt with the comments of the reviewers as follows: All the places which we changed have already been marked yellow in our paper, for the purpose of highlight. We hope the revised manuscript meets the standard of publication. Thank you!

Point 1: The experimental diet in phase 1 contained 20% lactose while 12.88% lactose was incorporated in the diet. What other sources of lactose were in the diet. I can imagine that it is the whey protein. If so, please mention the quality of both the lactose and the wheyprotein.

Response 1: Thank you very much for the comments and suggestions. It is whey protein. The mass content of lactose and whey protein is about 62.5% and 37.5% respectively. Finally, after comprehensive consideration by our team and referring to previously published articles, we decided to delete the lactose content parameter in the calculated value list.

Point 2: The authors do not motivate the choose for the days the fecal samples were taken. The majority of the diarrhea problems occur between 5 and 12 days postweaning. Yet, the digestibility samples and the samples for microbial analyses were obtained well after the critical period. They should motivate why they choose these periods. At day 35 and 42 p.w. the intestinal conditions are already stable. So one can expect that differences will be less than in the critical period.

Response 2: Thank you very much for the comments and suggestions. We chose the current experimental method due to reasons such as experimental design and enterprise cooperation project. For the places where the experiment is insufficient, we will fully consider your comments and suggestions on the follow-up experiment in the next experiment.

Point 3: The diets fed are "luxureous" and highly digestible. In order to measure an interaction a more risky diet should have been used. Even an effect of high levels of ZnO on diarrhea was not observed.

Response 3: Thank you very much for the comments and suggestions. Based on the results of this experiment, we feel that it is necessary to conduct further experimental research. We will refer to and adopt your suggestions and comments in subsequent experiments.

Point 4: In my opinion in the discussion the authors could have paid more attention between the relation ADG and G:F D22-42 and the digestibility data. The feed efficiency and the digestibility data seem to be correlated. The bacteriostatic effect of ZnO had an effect on both. The addition of ZnO causes that less andogeneous protein is formed what probably caused the hgiher protein digestibility and G:F.

Response 4: Thank you very much for the comments and suggestions. On the one hand, we felt that the discussion was not very clear. On the other hand, the team members believed that the main focus should be on the replacement of zinc oxide with probiotics, so we adopted the current discussion situation.

Point 5: I would not mention line 191-193.

Response 5: Thanks for your suggestion, we have edited it, please see line.

Round 2

Reviewer 1 Report

The manuscript is in the topic of “Evaluation on the growth performance, nutrient digestibility, faecal microbiota, noxious gas emission, and faecal score on weaning pigs supplement with and without probiotics complex supplementation in different level of zinc oxide”. I believe that results provided are important, methodology adequate and therefore this paper can be published in this Journal in present form.